# Intermittent cluster dynamics and temporal fractional diffusion in a bulk metallic glass

Birte Riechers [1], Amlan Das[2,3], Eric Dufresne [4], Peter M. Derlet [5] ✉ & Robert Maaß [1,2,6] ✉

Glassy solids evolve towards lower-energy structural states by physical aging. This can be characterized by structural relaxation times, the assessment of which is essential for understanding the glass' time-dependent property changes. Conducted over short times, a continuous increase of relaxation times with time is seen, suggesting a time-dependent dissipative transport mechanism. By focusing on micro-structural rearrangements at the atomic-scale, we demonstrate the emergence of sub-diffusive anomalous transport and therefore temporal fractional diffusion in a metallic glass, which we track via coherent x-ray scattering conducted over more than 300,000 s. At the longest probed decorrelation times, a transition from classical stretched exponential to a power-law behavior occurs, which in concert with atomistic simulations reveals collective and intermittent atomic motion. Our observations give a physical basis for classical stretched exponential relaxation behavior, uncover a new power-law governed collective transport regime for metallic glasses at long and practically relevant time-scales, and demonstrate a rich and highly non-monotonous aging response in a glassy solid, thereby challenging the common framework of homogeneous aging and atomic scale diffusion.

Predictable material properties guarantee safe applications, ensure reliability in service, and allow life-time assessments. Developing this predictability remains a foremost activity of material science, and becomes challenging when the material is far from equilibrium. This is the case for amorphous and glassy materials, which are used in large quantities in construction, optical applications, or energy conversion[1–5]. Indeed, glassy solids undergo a permanent structural evolution that is described by a hierarchy of characteristic relaxation times. Over long enough periods of time, this structural activity can lead to a significant deterioration of structural and functional material properties, such as a complete loss of toughness for the case of metallic glasses (MGs) considered here[6]. Quantitative understanding of this structural evolution bears the promise of novel atomic-scale structure-property relationships, from which performance and life-time enhancements of the amorphous solid are expected.

In contrast to crystalline materials, amorphous solids lack long-range order in their atomic arrangements, and can be obtained by rapid quenching of a liquid into the meta-equilibrium of a super-cooled liquid (SCL) and below the glass transition temperature $T_g$, where dynamical arrest occurs (Fig. 1a). At lower temperatures and longer time-scales, this highly out-of-equilibrium solid-state evolves structurally through subtle atomic rearrangements[7,8] and may eventually crystallize (Fig. 1a). Such 'physical aging' corresponds to a sampling of the multidimensional potential energy landscape (PEL) and leads to energetically more favorable structural motifs[9]. As a result, heterogeneous structural length-scales manifested as subtle spatial

[1]Federal Institute of Materials Research and Testing (BAM), Unter den Eichen 87, 12205 Berlin, Germany. [2]Department of Materials Science and Engineering, University of Illinois at Urbana-Champaign, Urbana, IL 61801, USA. [3]Cornell High Energy Synchrotron Source, Cornell University, 161 Synchrotron Drive, Ithaca, NY 14850, USA. [4]Advanced Photon Source, Argonne National Laboratory, 9700 South Cass Avenue, Argonne, IL 60439, USA. [5]Condensed Matter Theory Group, Paul-Scherrer-Institute, CH-5232 Villingen PSI, Switzerland. [6]Department of Materials Engineering, Technical University of Munich, 85748 Garchingen, Germany. ✉e-mail: peter.derlet@psi.ch; robert.maass@bam.de

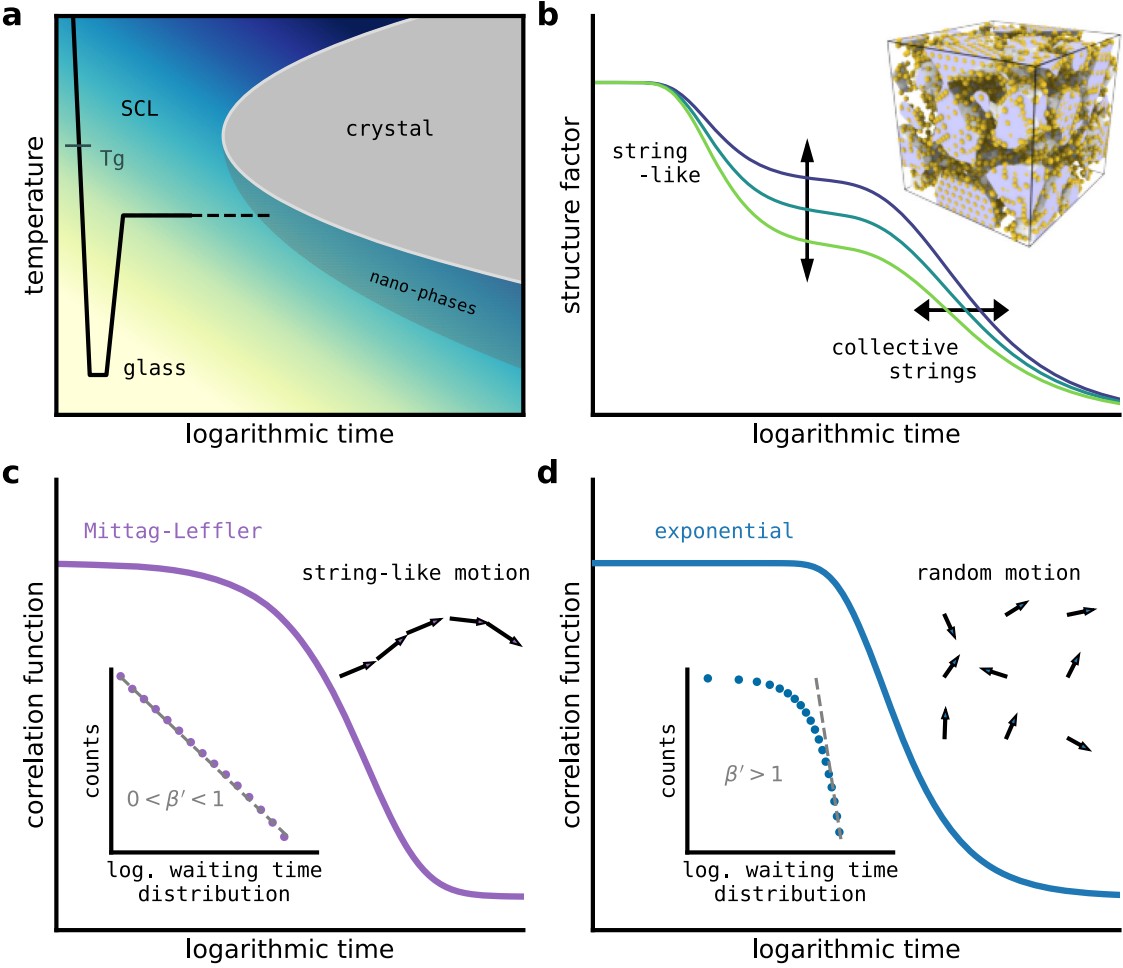

**Fig. 1 | Structural evolution and transport in a metallic glass. a** Time-temperature-transformation diagram with the indicated pathway from the super-cooled liquid (SCL) to a quenched glass, followed by a long-time anneal close to $T_g$. When the annealing protocol approaches the (non-)equilibrium nose, there will be an increased driving force towards low-energy motifs associated with crystal(-like) nano-phases. **b** The qualitative decorrelation picture emerging from atomistic simulation of a model binary glass, whose micro-structure is characterized by a percolating structure of low energy motifs (inset, gray iso-surface), referred to as the icosahedral-Frank-Kasper network for the model system considered, inter-dispersed with less relaxed domains. The faster decorrelation is mediated by fundamental microscopic changes in these excited domains, which over longer time-scales collectively decorrelate the percolating low-energy structure. **c** Stretched exponential/power-law (Mittag–Leffler) decorrelation due to correlated atomic motion (anomalous diffusion) where the average waiting time diverges. **d** Quantitative exponential decorrelation due to statistically independent atomic motion (normal diffusion), which in a solid is reflected in a distribution of waiting times with a convergent first moment. The insets in **c** and **d** represent double-logarithmic plots of the corresponding waiting-time distributions.

variations in density[10–12], elasticity[13,14] and time-scales[15,16] develop in MGs. The quantitative physical mechanism(s) underlying aging and to what material state such structural evolution asymptotes to, continue to remain open questions of fundamental glass science, the answering of which will form the required framework for optimized engineering applications.

Clearly, a detailed and long-time quantification of atomic-scale structural dynamics and its evolution is needed to address physical aging, which poses immense experimental challenges both in terms of resolution and duration. It is here that atomistic simulations of model alloy systems continue to give useful insights[17–19] of how a complex multi-component MG may locally reorganize during aging. For example, the binary Wahnström Lennard-Jones model alloy[20], which crystallizes into the C15 laves structure at high enough temperature[21,22], results in amorphous solids containing a disordered system-spanning network of low-energy and low-volume icosahedral and Frank-Kasper local atomic environments as the system relaxes at temperatures below $T_g$ (inset of Fig. 1b). Within such icosahedral-Frank-Kasper (IFK) regions, structural evolution at the simulation time-scale is rare. However in the non-IFK regions, thermally-driven structural

fluctuations are observable and take the form of string-like excitations. These involve a chain of atoms spanning a length-scale of several bond lengths that mediates a transport which is approximately conservative with respect to free-volume[23] and coordination, and over long-enough time-scales leads to structural changes of the IFK regions[24]. String-like excitations are not unique to the Wahnström potential and have been seen in a variety of amorphous and deeply undercooled systems[25–29].

In a real multi-component amorphous alloy, the low-energy structural motifs and corresponding excitations could be quite different and the time-scales for structural evolution much longer. Here, atomistic simulations give a qualitative message of spatial and temporal heterogeneity and entail a microscopic structural decorrelation that is both correlated and collective. In particular, the strong temporal heterogeneity spanning from the pico- to the micro-seconds in atomistic simulations nurtures the view of analogous variations at the experimental time-scale. With the advent of x-ray photon correlation spectroscopy (XPCS), this latter class of dynamics can now be measured via the intermediate scattering function that probes the time-scales associated with structural decorrelation (Fig. 1b)[30,31]. Specifically,

changes in temporal evolution of XPCS speckle patterns over extended time intervals reflect individual atomic-scale reorganizations within the illuminated volume, which together lead to a measurable characteristic structural decorrelation time-scale.

Following classical protocols to study glass relaxation[32–34], past XPCS work of the SCL and close to $T_g$ tracked such decorrelations[35,36], which phenomenologically are well-described by the Kohlrausch–Williams–Watts (KWW) function. This KWW-function captures correlated dynamics of compressed or stretched form via a time-scale $\tau$ and a shape exponent $\beta$: $f_{KWW} = \exp(-2(t/\tau)^\beta)$. Its non-exponential behavior is seemingly universally applicable to glass relaxation and especially suited for equilibrium ergodic system-transitions[37]. However, because of its empirical nature it remains unclear if the KWW-phenomenology adequately carries fundamental information of the underlying heterogeneous relaxation processes during physical aging of out-of-equilibrium systems, such as the here considered MGs.

Inspired by the ability to perform XPCS at unprecedented time-scales, as well as by results of atomistic simulations and its consequences for material transport, we reveal here a rich spectrum of short- and long-term relaxation dynamics in a MG, whose average relaxation behavior nevertheless can be well described by a generalization of the stretched exponential function. In doing so, we give a direct physical interpretation of the phenomenological KWW-function, as well as revealing asymptotic power-law decay in the structural decorrelation. Together these results give strong evidence that temporal fractional diffusion occurs in MGs, giving a robust framework for the quantitative understanding of physical aging at application-relevant time-scales in these amorphous alloys.

## Results

### Probing complex atomic motion during aging

A key technical development to capture atomic-scale dynamics with sub-Ångström accuracy over representative volume elements was the advancement of highly brilliant coherent 3rd-generation synchrotron x-ray sources[38]. This enabled the time-resolved analysis of intensity fluctuations in reciprocal space, which gives the intermediate scattering function and thus information on the time-scales associated with structural decorrelation[39]. Through intensity-intensity cross-correlations over sequentially in-time acquired x-ray speckle patterns, XPCS allows for the measurement of structural fluctuations at interatomic distances in condensed matter, that at sufficiently high time resolution is able to resolve in-time isolated dissipative events. This capability has now been leveraged for about 10 years to MGs, being typically applied to experimental time-windows that range between 1000 s and 10,000 s, with exceptionally long probing times reaching 60,000 s during isothermal annealing[40]. Taken as a whole and much in agreement with the temporal heterogeneity in atomistic simulations, the emerging literature base provides a surprising variety of relaxation signatures in MGs, including aging that obeys time-waiting time-temperature superposition[35,40–47], stationary[42,44] or temporarily accelerated[41,42] dynamics, and most prominently intermittent aging[16,42,48–50] that violates the aforementioned superposition principle. This richness in dynamical features raises the question if in fact structural energy minimization in MGs is accompanied by continuously increasing structural relaxation times, as a traditional view on physical aging would entail, or if momentary relaxation times always remain a temporal snapshot of a non-monotonously evolving ensemble?

To answer this question, we pursue here in-situ XPCS on a $Zr_{52.5}Ti_5Cu_{17.9}Ni_{14.6}Al_{10}$ MG during an isothermal anneal at 668 K (nominally at 0.98 $T_g$) for 300,000 s (83.3 h). Speckle-pattern intensity correlations at the first structural peak, corresponding to $q = 2.60$ to 2.66 Å$^{-1}$ (see Supplementary Fig. 1), were measured to construct

two-time correlation functions (TTCFs) according to

$$C(t_1, t_2) = \frac{\langle I(p, t_1)I(p, t_2)\rangle_p}{\langle I(p, t_1)\rangle_p \langle I(p, t_2)\rangle_p}, \quad (1)$$

where $I(p, t_i)$ is the intensity of a pixel $p$ at time $t_i$ and $\langle \cdot \rangle_p$ denotes an ensemble average performed on all pixels of a speckle pattern. We note that $C(t_1, t_2)$ sensitively depends on $q$ and probes therefore structurally well-isolated atomic motion. The here measured $q$-range is sensitive to the lower-end of the medium-range order (MRO) length-scale[51]. Figure 2 depicts $C(t_1, t_2)$ for the complete isotherm, where the diagonal trace at $C(t_i, t_i)$ has the highest contrast values. With increasing time-difference between speckle patterns along $C(t_i, t_i + \Delta t)$ the contrast decreases, signifying how the atomic structure gradually evolves at the probed structural length-scale. In fact, as summarized in Supplementary Fig. 5, complete decorrelation is observed, which means that deconfined (as opposed to confined) atomic-scale dynamics occurs during the entire isotherm – a regime in which long-time structural decorrelation is then intimately connected to asymptotic transport.

Prior to a quantitative time-dependent assessment of $C(t_1, t_2)$, a simple glance along the diagonal of Fig. 2 indicates strongly varying structural activity with time. Sub-panel a represents smooth and monotonous aging via increasingly slower decorrelations with time, whereas sub-panel b highlights constant decorrelation indicating steady-state structural activity. After 202,500 s of isothermal annealing, within the time resolution of 2.5 s, an abrupt change of $C(t_1, t_2)$ emerges (sub-panel c), signifying intermittent structural dynamics, and sub-panel d demonstrates clear acceleration of structural rearrangements. As seen from the lower panel of Fig. 2, none of these prominent features in the time domain can be attributed to absolute variations in the scattered intensity, $I_0(t)$. Thus, over the time period of 300,000 s at 0.98 $T_g$, the glassy solid exhibits virtually all of the so far reported dynamical signatures in one and the same system and

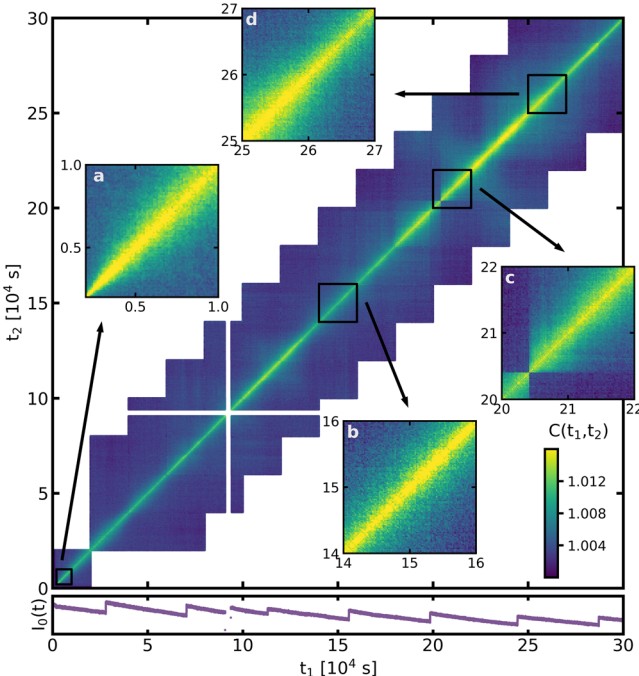

**Fig. 2 | Strongly varying decorrelation behavior over time.** Two-times correlation function, $C(t_1, t_2)$, of a $Zr_{52.5}Ti_5Cu_{17.9}Ni_{14.6}Al_{10}$ bulk MG at 0.98 $T_g$ measured by x-ray photon correlation spectroscopy in reflection mode. At approximately 95000 s a temporary x-ray beam loss led to the data gap. Various momentary decorrelation behaviors are highlighted with sub-panels a-d. Lower panel: scattered intensity, $I_0(t)$, as a function of annealing time.

experiment. Such a varied dynamical signature is much at odds with the expectation of gradually decreasing relaxation time-scales by homogeneous aging.

## Coupling across decorrelation time-domains

Quantifying $C(t_1, t_2)$ in terms of both its long-time decorrelation time and short-time component near the self-correlation time $C(t_i, t_i)$ reveals evidence of dynamics that are strongly influenced by structural reorganization at the sensitivity length-scale of the XPCS probe. Commonly, the time-scale of decorrelation is determined by fitting the data to the KWW-function. To reduce the noise of the data without losing finer details, pseudo one-time correlation functions, $g_2(t = t_1 - t_2, t_a = t_2) = \langle C(t, t_a) \rangle_{t_a}$, with $t_1 \geq t_2$ were determined from data sets ranging over $\Delta t_a = 1000$ s. After normalization by $R(t, t_a) = (g_2(t, t_a) - g_2(\infty, t_a))/(g_2(0, t_a) - g_2(\infty, t_a))$, the data was fitted to extract the time-scale of decorrelation, $\tau$, and the shape exponent, $\beta$. Further details on the related analysis are outlined in the Supplementary Information.

Alternatively to determining $\tau$ and $\beta$ from a KWW-fit, the first moment in time, $\langle t \rangle$, is calculated for each $\tau$-$\beta$-pair. Depicted in Fig. 3a, this quantification of a structural relaxation time-scale evolves very similarly as $\tau$ and returns time-scales between a few hundred up to 28,000 s. Overall, $\langle t \rangle$ follows a non-monotonous increase interspersed with significant fluctuations. As a function of waiting time, $t_a$, sub-time-domains can be identified that represent the earlier reported dynamical behavior, including monotonic aging ($0 < t_a < 25,000$ s; $132,000$ s $< t_a < 145,000$ s; $224,000$ s $< t_a < 250,000$ s), stationary dynamics ($60,000$ s $< t_a < 75,000$ s; $95,000$ s $< t_a < 110,000$ s; $150,000$ s $< t_a < 165,000$ s), or intermittent aging ($t_a = 205,000$ s), all highlighted in Fig. 3a. This demonstrates how the glassy solid pursues a non-trivial pathway of dynamical stages while exploring successive local minima and mega-basins of its PEL. Measurements over shorter and randomly selected time spans during aging may thus yield any of the

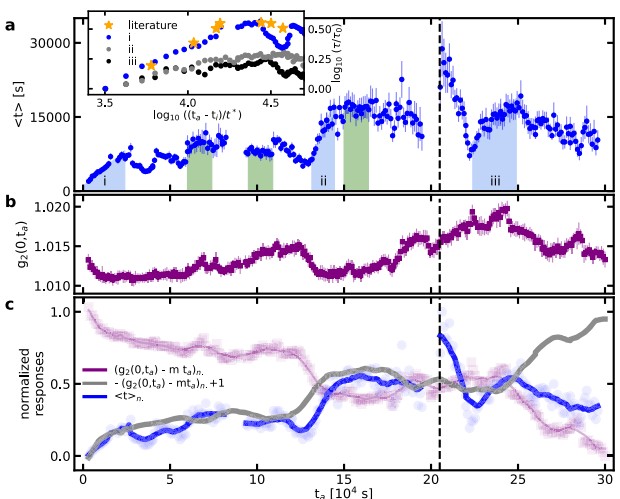

**Fig. 3 | Quantitative analysis of decorrelation behavior.** Time-scales of decorrelation, $\langle t \rangle$, and short-time plateau, $g_2(0, t_a)$, based on the two-time correlation function. **a** Time-scale based on $\beta$ and $\tau$ from a fit to a stretched-exponential function with an error bar of one standard deviation. Shaded areas mark sections of monotonic aging behavior (blue) and stationary dynamics (green). The inset compares the fitting parameter $\tau$ for the three observed regions of monotonous aging to literature data in Ref. 40. **b** Depicts the evolution of the contrast at the short-time plateau as a function of annealing time with standard deviation. In **c**, the correlation of $\langle t \rangle$ and $g_2(0, t_a)$ is visualized: Over the larger part of the experimental time window, these two quantities show an anti-correlation, revealed by subtracting a line of constant slope with $m = \frac{0.0165}{3 \cdot 10^5}$ s$^{-1}$ and flipping the data along a horizontal axis. Lines reflect smoothed data; symbols correspond to the original data point density. The vertical dashed black line across panels **a**–**c** marks a pronounced intermittent aging event.

momentary dynamics encompassed in Fig. 3a. Specifically, this can be demonstrated by considering the first 40,000 s of $\langle t \rangle$, during which a smooth increase is seen. For this time span, excellent agreement with earlier $T_g$-annealing of a Mg$_{65}$Cu$_{25}$Y$_{10}$ MG is found, and the continuously increasing dynamical time-scale can be rescaled and collapsed onto a single data set, supporting the empirically proposed time-aging time superposition[40], obeying $\tau(t, t_a) = \tau_0 \exp(t_a/t^*)$, where $\tau_0$ is the value of $\tau$ at $t_a = 0$ and $t^*$ is a fitting constant. The same approach can be applied to other selected time spans of $t_a$, including $130,000$ s $< t_a < 145,000$ s and $220,000$ s $< t_a < 250,000$ s (sections ii and iii in Fig. 3a and inset). The succession of specific dynamical phases of the aging structure culminates in an intermittent event observed at approximately 202,500 s (vertical dashed line in Fig. 3), identified by an abrupt decorrelation from one recorded speckle pattern to the next that leads to a marked slowing down of the structural activity. Indeed, $\langle t \rangle$ and $\tau$ (see Supplementary Fig. 6) double in response to the intermittent structural rearrangement. Shortly after, a subsequent and significant acceleration of the underlying atomic scale dynamics is seen and $\langle t \rangle$ attains values as low as in the early stages of physical aging.

Before addressing the underlying structural processes that may govern the highly complex dynamical evolution of the considered MG, we turn our attention to the short-time contrast, $g_2(0, t_a)$, shown in Fig. 3b. The time-dependent fluctuation of this measure contains all structural dynamics that is faster than the acquisition rate but that is not sufficiently fast to be averaged out fully, as would be the case for thermal vibrations. $g_2(0, t_a)$ therefore gives indirect access to the short-time component of the structural dynamics in the experiment. As seen in Fig. 3b, $g_2(0, t_a)$ exhibits a similarly erratic evolution over time as the long-time dynamical component depicted in Fig. 3a. Strikingly, several features as a function of $t_a$ coincide for both $\langle t \rangle$ and $g_2(0, t_a)$, but with a seemingly opposite trend. To evaluate this qualitative impression, $\langle t \rangle$ is plotted with $g_2(0, t_a)$ after subtracting a linear contribution, i.e., $g_2'(0, t_a) = g_2(0, t_a) - mt_a$ in a normalized representation. As can be deduced from Fig. 3c, the adjusted $g_2'(0, t_a)$ holds a strong anti-correlation with $\langle t \rangle$ for about 200,000 s. In other words, over an significantly long period of time there is, despite significant fluctuations, a non-monotonous overall slowing down of structural rearrangements represented by the long-time component $\langle t \rangle$, while the short-time component $g_2(0, t_a)$ of the data reduces in intensity. This robust anti-correlation prevails until the intermittent aging event at 202,500 s, after which for a few 10,000 s any link between both time-scale components is absent. However, at approximately 250,000 s a strong correlation between $\langle t \rangle$ and $g_2'(0, t_a)$ reemerges, which at least persists for the remaining 50,000 s - a time period that still largely exceeds the typical duration of XPCS experiments. As such, the data demonstrates a robust coupling between short and long-time intensity correlations that capture the atomic scale motion of the glassy solid.

## Evidence of intermittent cluster dynamics from atomistic simulation

The strongly non-monotonous evolution in both the short- and long-time domain, as well as its intermittent nature, raises the question which structural transport processes may dictate the complete intensity decorrelation of the speckle patterns. To bring light into this question, we pursue molecular dynamics (MD) simulations that, despite their time-scale difference with experiments, give a qualitative understanding of the unexpectedly diverse aging dynamics. We consider a well-relaxed Wahnström model binary glass consisting of a system-spanning IFK network, and perform isothermal simulations at 0.8 $T_g$ for several microseconds of physical time (see Supplementary Information for details) — a thermal regime that is expected to remain qualitatively valid at much longer time-scales and for significantly more relaxed and therefore extended structures at temperatures closer to $T_g$. In conjunction with these isothermal simulations we probe the structural evolution via TTCFs calculated from the simulated glass relaxation.

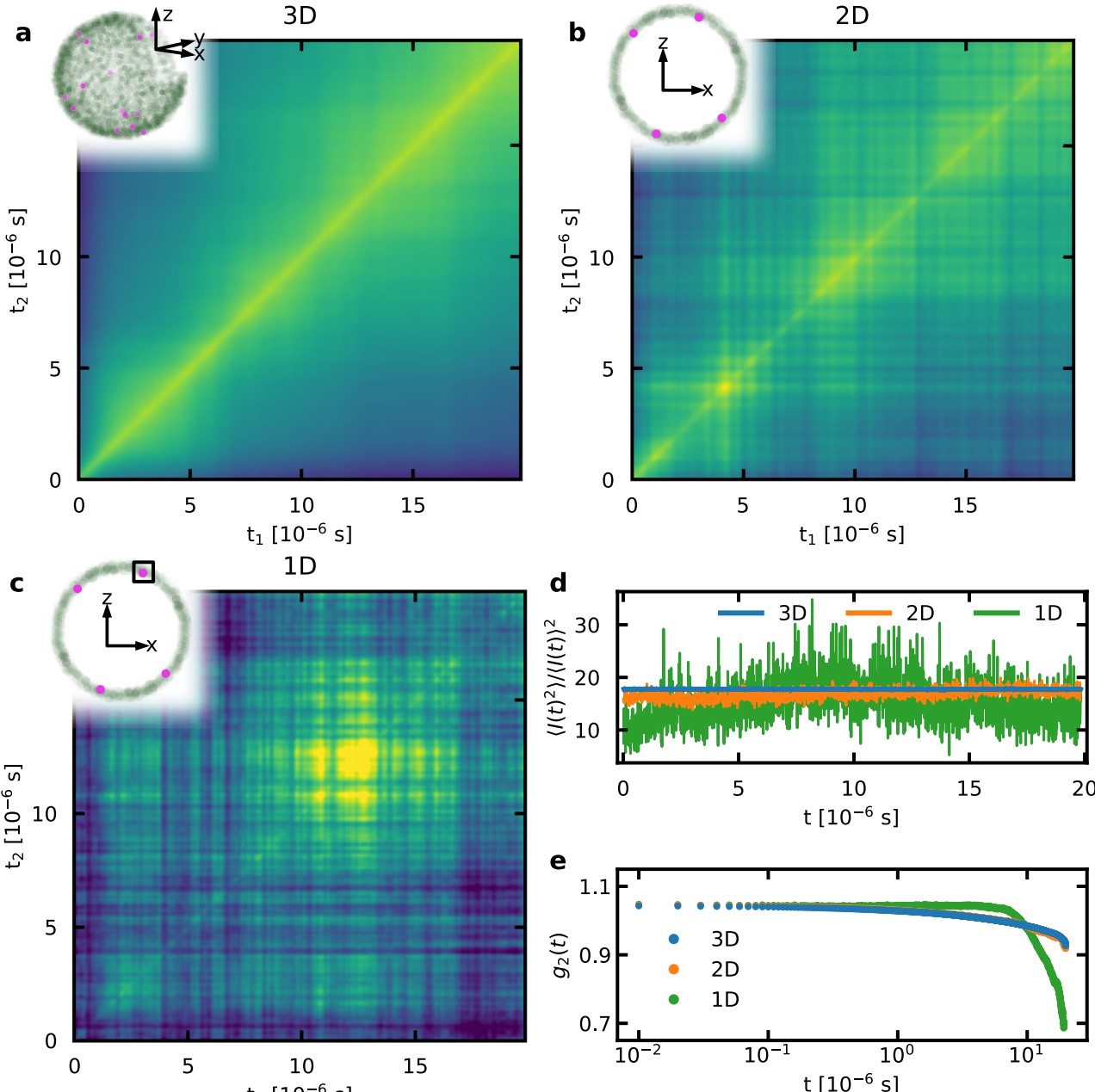

**Fig. 4 | Structural decorrelation behavior derived from simulated x-ray photon correlation spectroscopy.** MD-simulation data based on a well-relaxed Wahnström model binary glass under isothermal simulation conditions at 0.8 $T_g$ for ~20 μs of physical time. **a** Two-time correlation function (TTCF) based on the three-dimensional (3D) reciprocal space signal of the complete structure factor, **b** TTCF based on a two-dimensional (2D) cross-section of the complete structure factor, **c** TTCF based on a local one-dimensional (1D) spot with a width of $|\Delta \boldsymbol{q}| = 0.21\,\text{Å}^{-1}$.

Insets in panels **a**–**c** reflect the detected reciprocal space signal with increasing intensity reflected on a color-scale ranging from gray to pink. Panel **d** shows the auto-correlation $\langle I^2(t) \rangle / \langle I(t) \rangle^2$ corresponding to the contrast along the main diagonal of the TTCFs in panels **a**–**c**. Panel **e** states the one-time correlation function, $g_2(t)$, based on the individual TTCFs in panels **a**–**c**, normalized to values between 1 and 0.

Unlike in an XPCS experiment, which typically is restricted to a small region of reciprocal space by the detector ($p$ in Eq. (1)), the complete reciprocal space can be examined in simulations. Figure 4 displays the TTCFs originating from a) the entire three-dimensional (3D) structure peak, b) a two-dimensional (2D) cross-section of it, and c) a single segment (1D). The latter region spans a few pixels, each covering a $|\Delta \boldsymbol{q}|$ of 0.07 Å$^{-1}$ when scaled to the typical particle density, and is therefore comparable to the signal recorded experimentally on a detector plane. As a function of time, the speckle patterns will fluctuate due to changes in local structure that affect a particular pixel or pixels, and therefore structural length-scale, giving rise to the TTCFs

seen in Fig. 4. A similar situation is seen in the short-term dynamics contained in the self-correlation contrast that is summarized in Fig. 4d. In both components of the structural dynamics, it is seen that temporal fluctuations increase for smaller reciprocal space sampling, which demonstrates an increased sensitivity to structural changes affecting the particular choice of $\Delta \boldsymbol{q}$. Since this restricted choice of reciprocal space will be sensitive to a subset of structural changes at the MRO scale, they necessarily will be less frequent in time than the global average of local spatial changes. Specifically, the simulations reveal that abrupt collective adjustments of clusters of atoms can give rise to pixel-intensity fluctuations in a narrow $\boldsymbol{q}$-domain that manifest

themselves as the intermittent aging signature in the TTCFs. Using larger reciprocal space regions can therefore somewhat offset the small spatial volume being simulated. As in experiments, the one-time correlation function, $g_2(t) = \langle C(t, t_a) \rangle_{t_a}$ with $\Delta t_a$ equal to the full duration of the data set, can be determined from the temporal intensity-intensity correlation and is shown in Fig. 4e, demonstrating a convergence with respect to an increasing reciprocal space window. Despite the long simulated physical time of ~20 µs, overall decorrelation remains minimal and does not allow for robust fitting of a particular analytical form of $g_2(t)$.

Whilst it is difficult to measure the simulated diffusion constant resulting from multiple string-like excitations, one quantity easily obtainable is the waiting time for an atom to be displaced by a characteristic bond-length distance. Such a quantity can be measured irrespective of whether or not an atom experiences correlated or uncorrelated motion. The inset in Fig. 1c displays the obtained distribution of waiting times and demonstrates distinct "fat-tail"[52] power-law behavior $p(t) \sim (\tau/t)^{1+\beta'}$ with an exponent $1 + \beta' = 1.45$ that gives a divergent mean waiting time. This result is quite different from normal diffusion, which when viewed as a continuous time random walk, will have a finite temporal first moment $\langle \Delta t \rangle$ and finite spatial second moment $\langle |\Delta \mathbf{r}|^2 \rangle$, which together define the diffusion constant $D = \langle |\Delta \mathbf{r}|^2 \rangle / \langle \Delta t \rangle$. In turn, this defines the well-known asymptotic form of the mean-square displacement (Einstein) relation, $6Dt/2$, and the correlation function $g_2(t) \sim \exp(-|\mathbf{q}|^2 Dt)$ (see Fig. 1d) – relating structural decorrelation directly to the diffusion constant. Whilst $\langle |\Delta \mathbf{r}|^2 \rangle$ is expected to be well defined for both the solid and liquid phases, and of the order of typical nearest neighbor atom distance, the observed divergent $\langle \Delta t \rangle$ entails a simulated glassy transport mechanism not described by normal diffusion. We will soon see that this simulation inspired perspective allows for a new quantitative understanding of long-time temporal decorrelation measured experimentally with XPCS.

## Temporal fractional diffusion seen via XPCS

"Fat-tail" power laws often arise when unit-scale processes behave in a correlated and collective manner. The mathematical apparatus of fractional calculus[53] shows that when $0 \leq \beta' \leq 1$ for the waiting time-distribution, as seen in our simulations, the asymptotic mean-square-displacement becomes $6Kt^{\beta'}/[\beta' + 1]$ with $K = \langle |\mathbf{r}|^2 \rangle / \tau^{\beta'}$. This is sub-diffusive and therefore anomalous. Moreover, $g_2(t)$ should be given by $E_{\beta'}[-|\mathbf{q}|^2 Kt^{\beta'}]$, where $E_{\beta'}[\cdot]$ is the Mittag–Leffler (ML) function, which is plotted in Fig. 1c. Such diffusion is also referred to as temporal fractional diffusion, since the underlying diffusion equation is similar to Fick's first law but with a fractional temporal derivative[53,54], and has for example been used to understand anomalous dielectric spectra in glycerol[55]. Presently, this result yields a specific form for the correlation function $g_2(t)$ with a stretched exponential function for $t \ll \tau$, and with $E_{\beta'}[-(t/\tau)^{\beta'}] \sim \exp\left((t/\tau)^{\beta'}/[1+\beta']\right)$, whereas for $t \gg \tau$ a power-law decay dominates the decorrelation behavior with $E_{\beta'}[-(t/\tau)^{\beta'}] \sim \left((t/\tau)^{\beta'}/[1+\beta']\right)^{-1}$.

In the first instance, this gives a direct physical interpretation of the KKW-form for intermediate time-scales, where the so-called $\beta$-parameter is the exponent $\beta'$ defining the distribution of atomic displacement waiting-times. Deep in the glass, experiments often report $\beta > 1$[40,43,45,46], where this compressed decay is thought to arise due to quenched-in internal stress. If so, it is reasonable to assume that these stresses reduce over time even in the glassy state[35,56], so that $\beta < 1$ is observed at $T < T_g$ as in the here presented work. Indeed, in agreement with the above, Supplementary Fig. 6 reveals $\beta < 1$ over the entire measurement duration as in other well-relaxed MGs close to $T_g$[47,56]. On the other hand, asymptotically, the ML-function exhibits a power-law decay, which due to the short time windows probed until now has not been observed through XPCS measurements. In fact, fitting our intensity decorrelation over the very long experimental time window

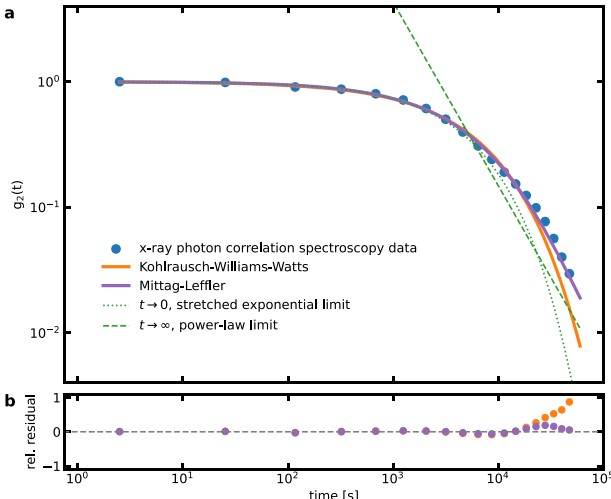

**Fig. 5 | Connecting decorrelation data to temporal fractional diffusion.** Mittag–Leffler (ML) and Kohlrausch–Williams–Watts (KWW) functions fitted to the one-time correlation function, $g_2(t)$, derived from the long-time x-ray photon correlation spectroscopy experiment. **a** Exponentially binned data of the one-time correlation function $g_2(t)$ (dots) normalized to values between 1 and 0, fitted with ML (purple line) and KWW (orange line) functions in double-logarithmic representation against time with indicators for the long-time limit of the ML-function by a power-law tail with exponent $\beta'$ (dashed line) and the short-time limit of the ML-function by a stretched exponential function (dotted line). At the largest decorrelation, the difference between the KWW-function and the ML-fitted datat amounts to ca. 9000 s. Panel **b** summarizes the relative residuals for fits to the ML- and the KWW-functions against logarithmic time. See Supplementary Information for the corresponding fitting results.

covered in the experiments, reveals that the ML-function more closely describes the more than four decades of time-scales covered by the synchrotron data (Fig. 5, see Supplementary Information for fitting details). Specifically, in the long-time domain of the power-law limit, the physics-based ML-approach clearly captures the experimental data with higher accuracy than the phenomenological KWW-fit. Whilst not easily discernible in Fig. 5, we note that the deviation between the KWW-fit and the ML-fitted data at the largest decorrelation is about 9000 s, which itself is a time amounting to the total experimental time of many previous XPCS experiments on MGs. We emphasize that the 300,000 s presented in Fig. 5 does not represent any physically relevant time scale. However, it is sufficiently long to uncover the here emerging power-law transport signature. As such, it gives a strong experimental evidence for temporal fractional diffusion in the studied MG and also demonstrates that the observed variety of heterogeneous structural dynamics is to be expected.

## Discussion

Isothermal annealing close to but below $T_g$ of a Zr-based MG reveals a fluctuating dynamical signature over a broad range of characteristic time-scales. The long decorrelation times as well as the strong fluctuations of all extracted parameters indicate continued structural relaxation, which is distinclty different to the short and stationary signature observed in the well-equilibrated SCL[36]. Since complete structural decorrelations are observed at all times, the underlying dynamics must be related to deconfined transport and therefore permanent atomic-scale evolution. Probed through intensity fluctuations of the structure-peak, it becomes apparent that the glassy solid explores non-monotonously its microscopic PEL, evolving towards some yet unknown structural asymptote. Probing such behavior for an unprecedented duration has revealed that anomalous diffusion mediates this activity and that the underlying atomic motion is collective, correlated, and intermittent. Simulation suggests the microscopic

mechanism of sequential displacements of a chain of atoms, resulting in transport and therefore structural decorrelation. What ever such thermally-activated microscopic activity might be for experimentally probed MGs, it is still expected to be localized, involving only a few atoms to achieve thermally-accessible activation energies of $O(eV)$.

These experimental and simulated observations need to be reconciled with the long history of classifying glassy structural evolution time-scales as slow $\alpha$-, and fast $\beta$- and $\gamma$-mechanisms, giving at least two characteristic time-scales[9,57–60]. The spatial heterogeneity of time-scales observed in simulation provides the essential link, suggesting an amorphous solid scenario for this terminology in which generic "fast" mechanisms mediate all structural fluctuations at the MRO length-scale ($\beta$-relaxation), whereas at much longer time-scales they collectively lead to fluctuations of the low-energy structural motif network ($\alpha$-relaxation). This MRO-scenario for $\beta$-relaxation differs from the short-range order viewpoint of "rattling-in-the-cage" within a liquid[61], whereas the $\alpha$-mechanism is now firmly linked to glassy micro-structural evolution. We propose that such a hierarchy of fluctuations, now giving a structural identification, is behind the diverse hetero-geneous temporal dynamics seen in the present long and short time-scale XPCS data. Moreover, because such micro-structure is system-spanning, the structural changes associated with $\alpha$-relaxation will mediate both the long-time elastic and plastic mechanical response − an example of which is the thermally-driven stress-biased structural dynamics seen in past XPCS work probing elastically loaded MGs[16].

Whilst the observation of heterogeneous temporal dynamics gives fundamental qualitative insight, the long-time average has also demonstrated quantitative understanding of the asymptotic transport properties − the power-law regime of anomalous diffusion. The iden-tified sub-diffusive behavior has been reported for colloidal glass transitions[62], but is quite different to super-diffusive transport of MGs in response to extremely high stresses[63]. In the latter, plastic defor-mation processes underlie the structural evolution and spatial fractional diffusion might occur. The here seen power-law regime of anomalous diffusion cannot be fully described by the KWW-approach, which exponentially suppresses long time-scale correlations. Our work does however show the conventionally applied KWW-form is well suited for the intermediate time-scale domain giving a physical inter-pretation to the $\beta$-exponent as the exponent of a waiting-time power-law distribution for correlated atomic-scale hopping.

Being able to capture time-scales ($\tau$, $\langle t \rangle$) primarily related to the collective emergence of fast MRO-processes that mediate aging via col-lective $\alpha$-activity, motivates continued exploitation of XPCS at the long-time extreme to establish the generality of the present fractional diffu-sion result. Alternatively, extending the XPCS technique to shorter times via developments in detector technology, would give increasingly direct insight into the fast microscopic processes of the $\gamma$- and $\beta$-mechanisms, and access to the entire temporal form of $g_2(0, t_a)$. This will (at least) have two plateaus: one occurring at times in which these fast processes are rare (the regime of atomistic simulations) and a second intermediate-time plateau occurring at times for which the $\gamma/\beta$-processes are fast, but not fast enough to collectively decorrelate the structural heterogeneities and therefore access the collective $\alpha$-mechanism of relaxation. Such a finding would be further experimental evidence for the proposed picture provided by atomistic simulations. Indeed, we anticipate that XPCS will be able to deliver the necessary quantitative experimental insight to connect the rich temporal and spatial microscopic dynamics in response to temperature, both of which underlie physical aging in MGs and meta-stable amorphous solids in general.

## Methods
### Sample preparation and thermal protocol
The MG sample investigated in this study is a bulk $Zr_{52.5}Ti_5 Cu_{17.9}Ni_{14.6}Al_{10}$ alloy (Vit 105, purchased from Liquidmetal) cut to dimensions of $4 \times 4 \times 1.5\,mm^3$ from plate material. To ensure stable positioning and low thermal lag throughout the annealing experiment, the sample was attached by Silver paste to a Titanium disc, which was mounted in an Anton Paar DHS 1100 heating chamber. The chamber was equipped with a Beryllium window transparent to the x-ray beam and was evacuated by a turbomolecular pump in advance to and throughout the annealing experiment. After heating the MG sample from $298\,K = 0.45\,T_g$ to $0.98\,T_g$, the temperature was held steady within a few mK throughout the entire measurement (see Supplementary Information for details).

### X-ray photon correlation spectroscopy
The XPCS experiments were performed at the 8ID-E beamline of the Advanced Photon Source at Argonne National Laboratory. The x-ray photon energy was set to 7.32 keV, corresponding to a wavelength of 1.69 Å. The scattering angle was ~1° in $2\theta$ around the maximum of the amorphous halo of the MG, corresponding to a $q$-range from $2.60\,Å^{-1}$ to $2.66\,Å^{-1}$. Based on the pair distribution function, $G(\mathbf{r}) = 4\pi r \left[\rho(\mathbf{r}) - \rho_0\right] = \frac{2}{\pi} \int_0^\infty \mathbf{q} \left[S(\mathbf{q}) - 1\right] \sin(\mathbf{qr})\, d\mathbf{q}$, the position of the first maximum in atomic density is related to $\mathbf{q}$ by a spherical Bessel-function $J_0(\mathbf{qr}) = \frac{\sin(\mathbf{qr})}{\mathbf{qr}}$, resulting in a probing of dynamics at intera-tomic distances around $\frac{5\pi}{2|\mathbf{q}|} \sim 3.0\,Å$ and therefore at the lower MRO length-scale[51]. The beam was focused onto the sample probing a volume of $10 \times 10 \times 25\,\mu m^3$ and was detected in reflection mode using an X-Spectrum Lambda 250 K Pixel Array Detector of size $19 \times 19\,mm^2$ (55 $\mu$m pixel size) at a distance of 1.1 m from the sample with a per-pixel resolution of $\Delta|\mathbf{q}| = 1.74 \times 10^{-4}\,Å^{-1}$. In total, $1.2 \times 10^5$ speckle patterns were recorded in non-top up mode with an acquisition time of 2.5 s. Applying a Matlab-routine developed at the APS beamline, all speckle patterns were cut with a pre-defined mask identical throughout the complete measurement, considering active and illuminated pixels only. Each pattern was corrected by the polynomial fit over the coarse-grained pixel-wise average over all patterns to account for the static diffraction background. The corrected speckle patterns were normal-ized by the ratio of average speckle pattern intensity and the average over all speckle pattern intensities, and serve as the basis for calcu-lating TTCFs.

### Analysis of two-time correlation functions
On-site analysis of the recorded speckle patterns was performed based on the Matlab-routines provided at 8ID-E. Along $1.6 \times 10^4$ adjacent frames, equivalent to a duration of $4 \times 10^4$ s, the TTCFs were calculated from the intensity, $I(p, t_i)$, of the recorded speckle patterns. Con-secutive TTCFs were stitched, and differences between the contrast signal of adjacent TTCFs are below the overall noise level of the con-trast data and do not influence further data treatment. Subsequently, the resulting data set was handled as one continuous data set during further analysis which was performed via python-based analysis rou-tines developed by the authors. The short-time plateau, corresponding to values along the main diagonal in the TTCF, was determined as the average of the four surrounding data points in the TTCF, $C(t, t+t')$, $C(t, t-t')$, $C(t+t', t)$, $C(t-t', t)$, where $t'$ is the measurement's tem-poral resolution of 2.5 s. The long-time plateau was identified by assuming that for the initial 140,000 s of the XPCS annealing experi-ment a complete decorrelation occurs, defining the long-time plateau at a contrast value of $g_2(\infty, t_a) = 1.00240(28)$. Before fitting of the data to KWW- and ML-functions, pseudo one-time correlation functions $g_2(t, t_a) = \langle C(t, t_a) \rangle_{t_a}$ with $\Delta t_a = 1000$ s were determined, which were binned to five points per decade to improve the signal-to-noise ratio, and normalized to the individual short-time plateau, $g_2(0, t_a)$, and the constant long-time plateau, $g_2(\infty, t_a)$, yielding $R(t, t_a)$. To construct the one-time correlation function, $g_2(t)$, shown in Fig. 5, the complete data set was averaged over $t_a$ and binned with an exponentially increasing bin-width.

## Data availability

The XPCS data analyzed in this study have been deposited at zenodo.org with the digital object identifier https://doi.org/10.5281/zenodo.12684513.

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

## Acknowledgements

B.R. gratefully acknowledges financial support by the German Aerospace Center (Grant No. 50WM2158), the European Union's Horizon Europe Framework Programme (HORIZON) under the Marie Skłodowska Curie grant agreement (No. 101063523), and BAM's Adolf Martens postdoctoral fellowship programme. The XPCS experiments were performed at the X-ray Science Division beamline 8ID-E of the Advanced Photon Source, a U.S. Department of Energy (DOE) Office of Science User Facility operated for the DOE Office of Science by Argonne National Laboratory under Contract No. DE-AC02-06CH11357.

## Author contributions

Conceptualization: R.M. Methodology: R.M. and P.M.D. Software: P.M.D. and B.R. Validation: B.R., A.D., E.D., P.M.D., and R.M. Formal analysis: B.R., A.D., and P.M.D. Investigation: A.D., and E.D. Resources: R.M. and P.M.D. Writing - original draft: B.R., P.M.D., and R.M. Writing - review and editing: B.R., A.D., E.D., P.M.D., and R.M. Visualization: B.R., and P.M.D. Supervision: R.M. Project administration: R.M. Funding acquisition: R.M. The authors thank T. Egami for critical discussions and valuable suggestions to the present work.

## Funding

## Competing interests

The authors declare no competing interests.
