## [Peer Review File · Nature Communications]

Intermittent cluster dynamics and temporal fractional diffusion in a bulk metallic glassEditorial Note: This manuscript has been previously reviewed at another journal that is not operating a transparent peer review scheme. This document only contains reviewer comments and rebuttal letters for versions considered at *Nature Communications*.

REVIEWER COMMENTS

I thank the authors for their patience and the detailed answer that has allowed me to follow better the work. Although I recognize the importance of the numerical results and the physical interpretation of the data, I still have new doubts each time that I read the work and I cannot support its publication in the present form.

1) I think that the article would benefit from a clearer presentation of the data

The authors call “decorrelation traces” the correlation curves extracted from the TTCFS averaged on 1000s. These curves are then normalized in the standard way. They then call “pseudo one curves $R(t, t_a)$ ” the average of 400 traces; and they call $g_2(t)$ the average of all traces. Being all these quantity intensity auto-correlation functions (more or less averaged), I don’t understand why the authors use this nomenclature which can create confusion. If all data represent the same observable with different time averaging, why simply don’t use $g_2(t)$ and $g(t, t_a)$ and explain it? What do the author mean with “pseudo” one curve? If it’s just for the dependence on the annealing time, it would be fixed. this would help the reader to avoid trying to find the differences.

2) To my point of view, the beautiful part of this work is the multi-dimensional reciprocal space analysis and the possibility to associate a physical mechanism to the data.

The authors strongly remind several time that the novelty is the long acquisition time and the fact that they observe many different kind of aging. As correctly stated by the first reviewer in the previous journal, this is not novel at all as it has been already observed in many previous studies. Notwithstanding they still use sentences like this one taken for instance from the abstract:

Our observations ..., and demonstrate a rich and highly non-monotonous aging response in a glassy solid, thereby challenging the common framework of homogeneous aging and atomic scale diffusion.

This was already demonstrated in previous works that are even cited in the article. I suggest again the authors to adjust their claim accordingly.

3) The response to aging close to T_g justifies the stretched decay of the functions, but I still think that the authors should clarify at the beginning of their work that not all XPCS data show stretched non-exponential decay as a reader would understand from pag. 5. Even in the excellent previous works of the same authors, the decays are compressed well below T_g , including those in the cited reference where the lowest value corresponds to a single exponential. From the results of the current work, it seems clear that the way aging progresses is independent on the KWW parameter. This is interesting but not commented at all in the work. I stress this point because stretched decay would imply a different motion than in previous low temperature works, and this is an important point that should be discussed in the manuscript.

4) Finally, concerning the ML vs KWW fitting. Although I have some preferences for the ML for its physical interpretation, I still don’t see any strong evidence to disclaim the KWW. On standard time scales, as shown in Fig S6 the two functions provide similar results. The only difference is observed in the curve obtained averaging all the 83h and only at very long times (basically after almost full decorrelation) in a log-log scale (while it would be invisible in linear-log). The authors claim that this is the relevant time scale that one should observe. My question is why 83h should be the most appropriate averaging of the data? We know that they would not find the same result averaging shorter times (Fig. S6). Would they find the same agreement waiting the double? Being the curve an average of randomly distributed regimes of relaxation, I don’t see how this could be confirmed. Averaging fluctuating data on temporal scales larger than the relevant scales of the fluctuations (Fig5) will inevitably contain less information than data with a better temporal resolution (Fig.S6). So I don’t see how this function should be more appropriate.

The authors don't need to smear out all the rich aging response by averaging the whole data set to support the use of the ML and the power law decay. The numerical results and the good quality of the analysis (Fig. S6) are in my opinion already enough to support their conclusions and prefer the ML to a pure phenomenological function used mainly for viscous liquids.

Reviewer Response Nature Communications Submission NCOMMS-24-04419A: Intermittent cluster dynamics and temporal fractional diffusion in a bulk metallic glass

Birte Riechers¹, Amlan Das², Eric Dufresne³, Peter M. Derlet^{*4}, and Robert Maaß^{†1,5}

¹Federal Institute of Materials Research and Testing (BAM), Unter den Eichen 87,
12205 Berlin, Germany

²Center for High-Energy X-ray Sciences, Cornell University, 161 Synchrotron Drive,
Ithaca, New York 14850, USA

³Advanced Photon Source, Argonne National Laboratory, 9700 South Cass Avenue,
Argonne, IL 60439, USA

⁴Condensed Matter Theory Group, Paul-Scherrer-Institute, CH-5232 Villigen PSI,
Switzerland

⁵Department of Materials Science and Engineering, University of Illinois at
Urbana-Champaign, Urbana, IL 61801, USA

April 16, 2024

Reviewer III

I thank the authors for their patience and the detailed answer that has allowed me to follow better the work. Although I recognize the importance of the numerical results and the physical interpretation of the data, I still have new doubts each time that I read the work and I cannot support its publication in the present form.

We thank the reviewer for the comments and questions but also for challenging the concepts presented in our manuscript, which certainly leads to improvements in the presentation of our findings. We also highly appreciate that we were able to convince the reviewer of the quality of our data and its interpretation.

We took the concerns raised by the reviewer in the last two evaluations seriously and adjusted both the manuscript and the supplementary materials (SM). In response to these two earlier revision rounds, we have added a very extensive description of both the analysis and the experiment amounting to 8 additional pages which give a considerable depth to the data and its analysis. Here, we continue our efforts and address in the following the four raised points.

1) I think that the article would benefit from a clearer presentation of the data

*peter.derlet@psi.ch

†robert.maass@bam.de

We agree with the reviewer that the representation of the data should be as clear as possible. From the here given comment by the reviewer, we realize that we have to improve the manuscript further, but at the same time it is our opinion that the previously submitted manuscript already gave a rather comprehensive picture. In our following response, we individually address the single points raised by the reviewer in comment 1).

The authors call “decorrelation traces” the correlation curves extracted from the TTCFS averaged on 1000s.

Unfortunately, when we searched through manuscript and SM, we were not able to make out any passages where this is stated explicitly or implicitly. The term “decorrelation trace(s)” appeared in the manuscript solely within the figure captions of Fig. 4 and Fig. 5, and in the SM, where “decorrelation traces” were defined in the Materials and Methods section (page 20, *Analysis of two-time correlation functions*) as “[...] momentary correlation functions $g_2(t, t_a)$ at full temporal resolution [...]”.

To settle this minor semantic issue, we decided to remove the phrase completely from the main manuscript, and to reduce its usage in the SM. Now, the first occurrence of this term is in the Supplementary Text in the paragraph *Treatment of decorrelation traces in preparation to fitting procedures* on page 24, which starts with the sentence: “In the following, the text refers to “decorrelation traces”, which are hereby defined as the data reflecting decorrelation along a horizontal line in the TTCF at full time resolution starting from the TTCF’s diagonal, thus corresponding to $g_2(t, t_a) = \langle C(t, t_a) \rangle_{\Delta t_a}$ with $\Delta t_a = 2.5$ s, i.e., the time resolution of the experiment.”

These curves are then normalized in the standard way. They then call “pseudo one curves $R(t, t_a)$ ” the average of 400 traces;

Again, we need to point out that our submitted manuscript and SM does not contain the phrase “pseudo one curves” anywhere in the text. We assume that the reviewer refers to the term “pseudo one-time correlation function”, which is defined on page 7 of the manuscript by “a pseudo one-time-correlation function, $g_2(t = t_1 - t_2, t_a = t_2) = \langle C(t_1, t_2) \rangle_{\Delta t_a}$, with $t_1 \geq t_2$ was determined from data sets ranging over $\Delta t_a = 1000$ s”. As such, irrespective if it is standard language or not, it is very clearly defined. Apart from one other usage of the term “pseudo one-time correlation function” on the same page, this term is not reoccurring in the main manuscript. It does appear again in the SM, where, for the reviewer’s and the readers’ convenience, we tried to simplify the text and its usage.

We emphasize, that the normalized response, $R(t, t_a)$, is defined in a clear and accurate fashion on page 7 of the manuscript, based on changes made during the previous revision round.

and they call $g_2(t)$ the average of all traces.

Indeed, we have not given a clear definition of $g_2(t)$ in the text and only described it in the captions of Figs. 4 and 5 by: “[...] states the one-time correlation function $g_2(t)$, corresponding to the average of all decorrelation traces starting from the main diagonal for the individual TTCFs [...]” While we think that this definition is still unambiguous and identical to the concept of $g_2(t)$ in other XPCS-related publications, we address this now with adding an expanded definition and adjustment of the captions of Figs. 4 and 5. Added on page 10 in the main manuscript it reads “[...] the one-time correlation function, $g_2(t) = \langle C(t, t_a) \rangle_{\Delta t_a}$ with Δt_a equal to the full duration of the data set [...]”.

Being all these quantity intensity auto-correlation functions (more or less averaged), I don't understand why the authors use this nomenclature which can create confusion. If all data represent the same observable with different time averaging, why simply don't use $g_2(t)$ and $g(t, t_a)$ and explain it? What do the author mean with "pseudo" one curve? If it's just for the dependence on the annealing time, it would be fixed. this would help the reader to avoid trying to find the differences.

We sincerely hope that by the answers given here and by the corresponding revisions described above, we address this first comment thoroughly and to the satisfaction of the reviewer and the editor.

2) To my point of view, the beautiful part of this work is the multi-dimensional reciprocal space analysis and the possibility to associate a physical mechanism to the data. The authors strongly remind several time that the novelty is the long acquisition time and the fact that they observe many different kind of aging. As correctly stated by the first reviewer in the previous journal, this is not novel at all as it has been already observed in many previous studies. Notwithstanding they still use sentences like this one taken for instance from the abstract:

"Our observations ..., and demonstrate a rich and highly non-monotonous aging response in a glassy solid, thereby challenging the common framework of homogeneous aging and atomic scale diffusion."

This was already demonstrated in previous works that are even cited in the article. I suggest again the authors to adjust their claim accordingly.

We thank the reviewer for the appreciation of our work in relation to the numerical data and our efforts to connect the decorrelation behavior invoked by structural relaxation to a physical mechanism.

We understand that the reviewer is of the view that we make claims on advances that had already been made by previous publications in the field. Further, the reviewer writes: "The authors strongly remind several time that the novelty is the long acquisition time and the fact that they observe many different kind of aging." We do not make novelty-claims apart from the aspect of the uncovering of "a new power-law governed collective transport regime for metallic glasses at long and practically relevant time-scales". We express, that the duration of the here presented XPCS-data is unprecedented, which is true and we see no evidence given by the reviewer that this is not the case. We indeed write that we see variations in the observed aging behavior, but we do not claim that this is novel, in contrary, we compare our findings in detail to those of the field. Overall, it is not clear to us which "claim" we supposedly make in our manuscript that would not be supported by facts. We support our view by stating all passages in the text that are (from our view and understanding) connected to the matter:

a) In the abstract, as partially already cited from the reviewer: "Our observations give a physical basis for classical stretched exponential relaxation behavior, uncover a new power-law governed collective transport regime for metallic glasses at long and practically relevant time-scales, and demonstrate a rich and highly non-monotonous aging response in a glassy solid, thereby challenging the common framework of homogeneous aging and atomic scale diffusion."

We do propose a power-law transport regime, we do reveal this via very long measurements of one and the same alloy, and we therefore highlight how the same glass undergoes a sequence of aging phases, single instance of which have indeed been published before. The latter is made clear in the manuscript, but at the same time is not the point. The point is that singular observations at shorter time scales for different alloys and different processing histories do not at all give the same physical understanding as if revealed and tracked for the same sample. The reviewer seems to totally ignore this point, as this has

been iterated numerous times before.

b) On page 5: "Inspired by the ability to perform XPCS at unprecedented time-scales, as well as by results of atomistic simulations and its consequences for material transport, we reveal here a rich spectrum of short- and long-term relaxation dynamics in an MG, whose average relaxation behavior nevertheless can be well described by a generalization of the stretched exponential function."

This is clearly found nowhere in the literature and a fully fact-based claim.

c) On page 5 and 6: "Taken as a whole and much in agreement with the temporal heterogeneity in atomistic simulations, the emerging literature base provides a surprising variety of relaxation signatures in MGs, including aging that obeys time-waiting time-temperature superposition [Ruta2012PRL, Ruta2013JCP, Wang2015ActaMat, Evenson2015PRL, Giordano2016NatComm, Gallino2018ActaMat, Lutich2018PRL, Kuchemann2018PRB, Amini2021PRM], stationary [Evenson2015PRL, Gallino2018ActaMat] or temporarily accelerated [Wang2015ActaMat, Evenson2015PRL] dynamics, and most prominently intermittent aging [Evenson2015PRL, Das2019NatComm, Das2020ActaMat, Xu2020JPCC] that violates the aforementioned superposition principle. This richness in dynamical features raises the question if in fact structural energy minimization in MGs is accompanied by continuously increasing structural relaxation times, as a traditional view on physical aging would entail, or if momentary relaxation times always remain a temporal snapshot of a non-monotonously evolving ensemble?"

Here we embrace and acknowledge earlier work, but also state what we show in our work. There are no unjustified claims.

d) On page 7: "Thus, over the time period of 300 000 s at $0.98 T_g$, the glassy solid exhibits virtually all of the so far reported dynamical signatures in one and the same system. Such a varied dynamical signature is much at odds with the expectation of gradually decreasing relaxation time-scales by homogeneous aging."

We underline for the general audience what the overarching difference to an established view is. The strong fluctuations in both short and long-time parameters across the full isotherm demonstrate this.

e) On page 8: "This demonstrates how the glassy solid pursues a non-trivial pathway of dynamical stages while exploring successive local minima and mega-basins of its PEL. Measurements over shorter and randomly selected time spans during aging may thus yield any of the momentary dynamics encompassed in Fig. 3A."

This is simply a fact.

From these sections in our manuscript it is evident that

- we clearly acknowledge the variety of XPCS-connected aging responses found in literature and compare our data to these. (cf. point c)

- we state that we present an unprecedentedly long XPCS-measurement in our manuscript that reflects the variety of aging responses found in literature. Both literature data and the here-presented data support each other, while the long measurement duration sets individual data sets into context. (cf. points a, b, c, d, e)

- we write that we challenge the concept of homogeneous aging for the here discussed metallic glasses in view of the above mentioned variety of aging responses. This has been done by previous XPCS-related literature on metallic glasses and polymers as well, ever since the emergence of intermittent dynamics. We present a clear advancement and it is well communicated that such observations have so-far not been well understood or explained in context to a microstructural picture. Therefore it is an unsettled topic in the glass and XPCS-communities. Homogeneous aging is a common concept whenever the aging response of an ensemble average is discussed, but thanks to the insights into the momentary aging behavior offered by XPCS, we are actually able to discuss the aspects connected to a rather heterogeneous dynamical and also structural picture. (cf points a and e)

If the editor or the reviewer find at any point in the text that we make claims in this context that we do not fulfill, we ask that such claims are clearly indicated to us - by direct citation and not by vague rephrasing based on personal interpretation. At this point we are convinced that we do not make unjustified claims, neither intentionally nor by mistake.

3) The response to aging close to T_g justifies the stretched decay of the functions, but I still think that the authors should clarify at the beginning of their work that not all XPCS data show stretched nonexponential decay as a reader would understand from pag. 5. Even in the excellent previous works of the same authors, the decays are compressed well below T_g , including those in the cited reference where the lowest value corresponds to a single exponential. From the results of the current work, it seems clear that the way aging progresses is independent on the KWW parameter. This is interesting but not commented at all in the work. I stress this point because stretched decay would imply a different motion than in previous low temperature works, and this is an important point that should be discussed in the manuscript.

As the reviewer suggests, we do now explicitly acknowledge that XPCS decorrelation functions can be fitted by compressed or stretched exponential functions on page 5.

Further, we added a sentence on page 12 citing literature that shows compressed decay functions: "Deep in the glass, experiments often report $\beta > 1$ [Ruta2013JCP, Giordano2016NatComm, Lutich2018PRL, Kuchemann2018PRB], where this compressed decay is thought to arise due to quenched-in internal stress. If so, it is reasonable to assume that these stresses reduce over time even in the glassy state [Ruta2012PRL, Soriano2021JPCM], so that $\beta < 1$ is observed at $T < T_g$ as in the here presented work."

4) Finally, concerning the ML vs KWW fitting. Although I have some preferences for the ML for its physical interpretation, I still don't see any strong evidence to disclaim the KWW.

Our work certainly does not disclaim the KKW form, in fact rather the opposite is achieved. Via the ML formalism and its asymptotic properties, we give a firm theoretical foundation to its use in the literature. However, it also implies that if one has access to longer times, an asymptotic power law dependence emerges, which is presented as one of the main points in the manuscript.

On standard time scales, as shown in Fig S6 the two functions provide similar results. The only difference is observed in the curve obtained averaging all the 83h and only at very long times (basically after almost full decorrelation) in a log-log scale (while it would be invisible in linear-log). The authors claim that this is the relevant time scale that one should observe. My question is why 83h should be the most appropriate averaging of the data? We know that they would not find the same result averaging shorter times (Fig. S6). Would they find the same agreement waiting the double?

We do not say that 83 hours is the most appropriate time scale. However, our findings do show how a

long measurement reveals new insights beyond earlier short-time assessments. To most accurately probe the identified regime, long times are required if one wants to probe asymptotic behaviour. Clearly, as much averaging as possible is desired. Using all the data is therefore justified, given the varied behaviour over shorter timescales. Limiting it to a given number of hours is beyond the point. In our revised manuscript, we articulate this on page 13 by adding the sentence: "We emphasize that the overall aging duration exceeding 300 000 s and used to construct the data presented in Fig. 5 does not represent any physically relevant absolute time scale. However, it is sufficiently long to uncover the here emerging power-law transport signature."

Being the curve an average of randomly distributed regimes of relaxation, I don't see how this could be confirmed. Averaging fluctuating data on temporal scales larger than the relevant scales of the fluctuations (Fig5) will inevitably contain less information than data with a better temporal resolution (Fig.S6). So I don't see how this function should be more appropriate.

Following the previous point, we are of the opinion that the many regimes of behaviour seen are the underlying fluctuations that contribute to the asymptotic (long time) behaviour, and therefore one must average over them. If one wants to look at the zoology of non-asymptotic behaviour, one can, but it is only through such averaging, that the power-law signature will become manifest.

The authors don't need to smear out all the rich aging response by averaging the whole data set to support the use of the ML and the power law decay. The numerical results and the good quality of the analysis (Fig. S6) are in my opinion already enough to support their conclusions and prefer the ML to a pure phenomenological function used mainly for viscous liquids.

We thank the reviewer for this positive final statement, as it tells us that he/she does see a value in how we have presented our findings. In other words, we absolutely agree on this comment that firmly condenses what our manuscript presents.

REVIEWERS' COMMENTS

Reviewer #3 (Remarks to the Author):

I thank the authors for their patience in addressing all my concerns.
I believe this work will have a major impact in the community. I therefore highly recommend its publication